# The Retinal Dopaminergic Circuit as a Biomarker for Huntington’s and Alzheimer’s Diseases

**DOI:** 10.3390/ijms26125532

**Published:** 2025-06-10

**Authors:** Pedro Blanco-Hernán, Lorena Aguado, María José Asensio, Ana Gómez-Soria, Pedro de la Villa, María José Casarejos, Alicia Mansilla

**Affiliations:** 1Instituto Ramón y Cajal de Investigación Sanitaria (IRYCIS), 28034 Madrid, Spain; 2Servicio de Neurobiología-Investigación, Hospital Ramón y Cajal, 28034 Madrid, Spain; 3Department of Systems Biology, Universidad de Alcalá (UAH), 28871 Alcalá de Henares, Spain

**Keywords:** retina, Huntington’s disease, Alzheimer’s disease, dopaminergic circuit, colour contrast

## Abstract

Retinal dysfunction is emerging as a potential early marker of neurodegenerative diseases. Within the retina, the dopaminergic circuit, comprising dopaminergic amacrine cells, dopamine synthesis and turnover, and dopamine receptor signalling, is essential for visual processing, particularly colour contrast perception. Disruption of this circuit may underline early retinal alterations observed in Huntington’s disease (HD) and Alzheimer’s disease (AD). In this study, we systematically analysed retinal dopaminergic dysfunction in murine models of HD (genetic origin) and AD (sporadic), across different disease stages. We assessed dopamine levels, turnover, tyrosine hydroxylase expression, D1 and D2 receptor gene expression, and neurotransmitter balance. HD mice showed early and marked alterations: reduced dopamine content, decreased tyrosine hydroxylase, increased turnover, and downregulation of D1 receptor expression—all preceding motor symptoms and detectable brain pathology. In contrast, AD mice showed only mild changes at later stages; however, clinical evidence suggests that similar dysfunction may occur earlier in human AD. These findings position retinal dopaminergic disruption as a potential early biomarker in HD and possibly in AD. While the current study relies on invasive techniques in animal models, it lays the groundwork for non-invasive retinal assessments, such as electroretinography or optical coherence tomography, as promising tools for early diagnosis and disease monitoring in neurodegeneration.

## 1. Introduction

Early diagnosis of neurodegenerative diseases is essential to enable timely intervention and improve clinical outcomes. A major drawback is that symptoms usually appear after substantial neuronal loss, significantly narrowing the window for early intervention [1]. Moreover, current diagnostic approaches are often invasive, costly, or impractical for large-scale screening. This has led to growing interest in peripheral biomarkers that can reflect early neurodegenerative changes. The retina offers a unique and accessible window into the central nervous system, as it can be evaluated non-invasively through imaging and electrophysiological techniques such as optical coherence tomography (OCT) and electroretinography (ERG) [2,3]. OCT allows a precise measurement of retinal layer thickness, providing structural insights into disease progression, while ERG evaluates the retina functional integrity by measuring the electrical responses to light stimulation [4]. The ERG a-wave reflects the activity of photoreceptors (rods and cones), and the b-wave is generated by bipolar cells, indicating inner retinal function [4].

In this study, we focused on the retinal dopaminergic circuit (RDC), which plays a key role in visual processing and contrast sensitivity and may be particularly vulnerable in the early stages of neurodegeneration. To investigate this, we used murine models of Huntington’s disease (HD), a genetically inherited condition, and Alzheimer’s disease (AD), which typically has a sporadic onset. These models represent two distinct pathological contexts, both of which involve dopaminergic system alterations. Although our experimental approach relied on invasive techniques limited to animal models, our objective was to characterize early RDC dysfunction that may ultimately support the development of non-invasive diagnostic tools in humans, such as ERG or OCT.

To frame our hypothesis, we begin by briefly reviewing the existing literature on retinal abnormalities observed in HD and AD, which collectively suggest a potential contribution of the dopaminergic circuit to disease-related retinal changes.

### 1.1. Retinal Pathology in Huntington’s Disease Patients

HD is a rare neurodegenerative disorder affecting 5–10 per 100,000 individuals, marked by involuntary movements, cognitive decline, and psychiatric symptoms like depression and apathy, typically beginning around age 35 [5]. HD results from an expanded CAG trinucleotide repeat in exon 1 of the huntingtin gene, producing a mutant huntingtin protein (mHTT) that loses normal function and gains neurotoxic properties. Medium spiny neurons (MSNs) in the caudate nucleus, especially those expressing dopamine receptor 2 (D2R), are most affected, though D1R-expressing MSNs are also impacted but more resistant [6].

Retinal changes in HD remain unclear, likely due to small sample sizes. OCT studies have shown increased outer nuclear layer (ONL) thickness, reduced macular volume, and thinning of retinal nerve fibre layer (RNFL) [7,8,9,10,11]. Cone-mediated circuits support high-acuity and colour vision in bright light, while rods are specialized for low-light (scotopic) conditions. In HD, studies report increased photopic activity [12], along with reduced multifocal ERG (mfERG) responses (reflecting macular dysfunction) and decreased full-field ERG (ffERG) signals, indicating global retinal impairment [13]. Meanwhile patients show visual deficits, such as colour and form contrast deficits [8,10,14,15]. All the findings in the HD retinal pathology in humans have been summarized in Appendix A.

### 1.2. Retinal Alterations in Mice Models of Huntington’s Disease

In HD mice models, the main retinal hallmark is mHTT accumulation across retinal layers, a phenomenon not yet observed in humans, except for one case without detectable deposits [16]. In mice, mHTT appears in retinal ganglion cells (RGCs) by 7 weeks and spreads to all layers by 11 weeks. Photoreceptors, particularly cones, are primarily affected, and its synapsis with bipolar cells exhibits abnormalities [17,18,19]. However, inner retina damage remains unclear. The RNFL shows thinning, and optic nerve structural alterations are observed [11,20,21].

HD pathology is characterized by a disruption in the circadian rhythm and sleep disturbances. These processes are regulated by the suprachiasmatic nuclei (SCN), which receive input from intrinsically photosensitive RGCs (ipRGCs) [22]. Besides circadian rhythm, these melanopsin-expressing ipRGCs regulate non-visual functions like pupil diameter. In HD mice, ipRGCs begin to degenerate at 7 weeks, leading to impaired pupil responses [23,24].

A decline in visual acuity is detected in 24-week-old mice [17]. Findings on ERG dysfunction in HD models are inconsistent. Several studies report reduced ERG responses under photopic conditions as early as 6 weeks of age [19,25,26]. Ragauskas et al. observed a decline in the b-wave at 8 weeks, with complete loss of the pattern ERG (pERG), and in two of six mice, a total absence of the b-wave [27]. In contrast, another study reports no significant ERG alterations until 16 weeks [17]. Retinal alterations in HD mice are summarized in Appendix A.

### 1.3. The Retinal Pathology in Alzheimer’s Disease Patients

Alzheimer’s disease (AD) is a chronic and devastating neurodegenerative disorder and the leading cause of dementia worldwide. It affects 11% of people over 65 in the U.S. [28,29], with prevalence expected to triple by 2050 [2,30]. The disease progresses from mild cognitive impairment (MCI) to severe memory loss, behavioural disturbances, and impaired communication, ultimately leading to complete loss of independence. AD is characterized by the accumulation of amyloid β-protein (Aβ) plaques and neurofibrillary tangles containing hyperphosphorylated tau (p-Tau) [31,32].

In AD, RGCs (including ipRGCs) are the most affected, with Aβ plaques concentrated in this layer [33]. On the other hand, p-Tau inclusions are found in all retinal layers [34].

Some OCT studies show that the ganglion cell layer (GCL) and inner plexiform layer (IPL) become thinner in AD patients, and this thinning is linked to cognitive decline. However, patients with the APOε4 gene (a risk factor for AD) show the opposite pattern [35,36,37]. In AD, the nerve fibre layer around the optic nerve and macular volume are often reduced [38,39,40,41,42,43,44,45,46]. Still, other studies report inconsistent connections with cognitive decline [47].

The pERG is decreased in AD patients [48,49]. Multifocal ERG waves are reduced and correlate with RNFL thinning as measured by OCT, reinforcing OCT’s potential for early AD diagnosis [50,51,52]. AD patients also show spatial and colour contrast deficits linked to macular atrophy and RNFL loss. Kim et al. found poor Ishihara’s colour test scores in AD patients, highlighting its potential as a diagnostic tool [53].

All the findings shown are reviewed in Appendix A.

### 1.4. The Retinal Pathology in Alzheimer’s Mice Models

The use of Curcumin-Aβ complexes in vivo show retina Aβ depositions appear from 12-weeks, before the onset of cognitive symptoms [54,55]. The Aβ deposits are more numerous in the inner layers of the retina, being highly localized in the GCL [56,57,58]. As suggested by the human AD pathology, mice strains confirm RGCs as the most vulnerable cells in the AD retina [57,59,60,61]. But in contrast to humans, no changes in ipRGC have been found [57].

OCT findings support the histological discoveries, showing GCL, ONL, and RNFL to be thinner than wild-type counterparts at 12 weeks [57,59,61].

ERG studies yield conflicting results. One study reports no changes in ffERG but detected RGC dysfunction via multielectrode array recordings in 16-week-old mice [62]. Conversely, other studies demonstrate altered ERG responses, including increased scotopic a-wave amplitude in 12-week-old mice [63], although other authors have found that the scotopic a-wave amplitude is reduced [64]. Additionally, rods exhibit impaired function, as evidenced by increased scotopic a-wave latency and decreased a-wave and b-wave amplitudes [65,66].

AD mice also show visual symptoms, such as reduced visual acuity and colour contrast deficit [62,67]. Vi et al. built a visual-stimuli four-arm maze that permitted the analysis of colour contrast deficit in mice. They observed a gradual decline in colour contrast starting at 34 weeks of age.

All findings from AD mice retinal pathology are summarized in Appendix A.

### 1.5. The Retinal Dopaminergic Circuit

In both HD and AD, three key retinal alterations have been reported: impaired colour contrast, thinning of the retinal nerve fibre layer (RNFL), and early changes in ERG responses. These findings point to dysfunction in the photopic pathway—comprised of cones, ON/OFF cone bipolar cells, and retinal ganglion cells (RGCs) which depend on dopamine (DA) signalling.

This dopaminergic signalling originates from a small population of tyrosine hydroxylase (TH)-expressing amacrine cells (DACs), known to regulate colour and spatial contrast, as well as light sensitivity. DACs receive excitatory input from ON bipolar cells and intrinsically photosensitive RGCs (ipRGCs), and their DA release is light-dependent, peaking under high illumination [68,69]. These functions make the RDC a modulatory circuit that adapts the retina to photopic conditions.

DA receptors are widely expressed in the retina. Photoreceptors express D4R, which modulates calcium channels and cone/rod gap junction coupling. Cone bipolar cells predominantly express D1R, enhancing light sensitivity and contrast perception. RGCs express both D1R and D2R; D1R improves contrast sensitivity, while D2R’s function remains unclear [68,69]. DACs express D2R as an autoreceptor to regulate DA release [68,70] (Figure 1).

Based on findings in both human and animal models, the retinal dopaminergic circuit (RDC) appears to be an early and vulnerable target in neurodegeneration. Here, we present the first detailed characterization of RDC dysfunction in HD and AD models, underscoring its potential as a disease-specific biomarker. Its involvement in colour contrast and light adaptation makes it a promising target for early detection strategies.

## 2. Results

### 2.1. Dopaminergic Circuit Alterations in the Retinas of HD Mice

Previous studies in murine models and human patients have identified alterations in the photopic visual pathway, where dopamine (DA) plays a crucial role in modulating light sensitivity, colour contrast, and overall visual processing. Given the importance of DA in retinal function, we quantified its levels in a mouse model of HD using high-performance liquid chromatography (HPLC) at two time points: at 10 weeks of age, before the onset of motor signs, and at 32 weeks of age, when motor symptoms are fully expressed [71].

Our results reveal a significant reduction in DA levels in HD mice as early as 10 weeks of age, indicating an early onset of dopaminergic dysfunction in the retina (Figure 2A). Note that DA levels in the retina are reduced at 32 weeks compared to 10 weeks.

Since tyrosine hydroxylase (TH) is the rate-limiting enzyme in DA synthesis, its expression serves as an indirect marker of dopaminergic neuronal activity. We quantified TH protein levels using western blotting. The results demonstrated a significant reduction in TH expression in HD mice at both points analysed (Figure 2B) and Appendix A, which is consistent with the observed decrease in DA levels, suggesting impaired function or loss of dopaminergic cells.

To further investigate DA metabolism, we measured DA degradation metabolites, homovanillic acid (HVA) and 3,4-dihydroxyphenylacetic acid (DOPAC). The DOPAC+HVA/DA ratio was calculated as an index of DA turnover, providing insight into the rate of DA degradation relative to its availability. DA turnover showed an increase only in 32-week-old HD mice (Figure 2C) where an increase in HVA but not in DOPAC is detected (Appendix A). Normally, DA is metabolized into DOPAC via MAO (monoamine oxidase) and then further converted into HVA by COMT (catechol-O-methyltransferase). An increase in HVA levels without a corresponding rise in DOPAC suggests enhanced COMT activity, potentially influenced by oxidative stress or other physiological factors affecting DA catabolism.

To gain a broader understanding of alterations in DA signalling beyond dopaminergic cell activity, we further assessed the expression of dopamine receptors (D1R and D2R) at the transcriptional level. Using quantitative real-time PCR (qRT-PCR), we found a statistically significant reduction in D1R mRNA expression in HD mice at both ages examined, indicating a dysregulation in dopamine-mediated signalling (Figure 3). D2R expression in HD mice tended to be reduced compared to controls, although the difference was not statistically significant.

These findings collectively indicate that HD mice exhibit profound alterations in the retinal dopaminergic circuit, including reduced DA synthesis, increased DA degradation at later disease stages, and dysregulated DA receptor expression, all of which may contribute to the progressive visual deficits observed in this disease model.

### 2.2. Dopaminergic Circuit in AD Mice Retinas

To gain a deeper understanding of whether the alterations observed in the retinas of HD model mice, particularly concerning the dopaminergic system, are specific to this model or represent a broader characteristic of neurodegenerative processes, analogous studies were conducted using an Alzheimer’s disease (AD) mouse model. These investigations were performed at two distinct age intervals. The first group consists of 27-week-old mice, a stage where no apparent degenerative phenotypes are observed. The second group includes 64-week-old mice, an age associated with the initial signs of degeneration.

Given DA’s key role in photopic visual pathways, which are impaired in AD patients and mouse models, we measured DA levels in AD mice using HPLC. At 27 weeks, levels were comparable to WT, suggesting intact early dopaminergic function. However, a significant decline at 64 weeks (Figure 4A) indicated progressive dysfunction with disease progression.

To further elucidate the dynamics of the dopaminergic system, we conducted western blot analyses to measure the expression levels of tyrosine hydroxylase (TH), the rate-limiting enzyme in DA synthesis, in both age groups. At 27 weeks, TH levels were similar between AD and WT mice. However, at 64 weeks, TH levels significantly increased despite reduced DA levels (Figure 4B and Appendix A), suggesting a compensatory response to DA depletion by upregulating TH to enhance precursor availability for DA synthesis.

Reduced DA levels may stem from increased degradation or impaired synthesis, though the latter seems unlikely given the TH levels. To investigate further, we assessed DA turnover by calculating the (DOPAC+HVA)/DA ratio. At 27 weeks, AD mice showed a lower ratio than WT, driven by reduced DOPAC and HVA levels (Appendix A), but at 64 weeks, the ratio was slightly higher.

To obtain a comprehensive understanding of DA signalling alterations in the AD mouse model, we analysed the gene expression levels of dopamine receptors. Specifically, the expression of D1R and D2R was quantified using qRT-PCR. D1R expression remained unchanged in 27- and 64-week-old AD mice compared to WT (Figure 5A), suggesting it is not significantly affected by disease progression. In contrast, a significant increase in D2R gene expression was observed exclusively in the 64-week-old AD mice, while no alterations were detected in the younger cohort (Figure 5B). This observed increase in D2R expression may reflect an adaptive attempt to enhance dopaminergic sensitivity and maintain neurotransmission efficiency despite the progressive loss of DA availability.

### 2.3. Other Neurotransmitters in HD Retinas

The balance between excitatory and inhibitory neurotransmitters is essential for accurate retinal signal processing. Glutamate (GLU), the primary excitatory neurotransmitter, mediates vertical transmission from photoreceptors to bipolar and ganglion cells, while GABA and glycine (GLY) provide inhibitory modulation through horizontal and amacrine cells, refining visual input and enhancing contrast resolution. Disruption of this excitatory–inhibitory equilibrium can impair light adaptation, contrast sensitivity, and overall visual function. Although this study focuses on the dopaminergic circuit, analysing GLU, GABA, and GLY is relevant because dopaminergic amacrine cells interact with both excitatory and inhibitory interneurons. Thus, RDC dysfunction could influence the overall neurotransmitter balance in the retina, contributing to broader functional impairments.

In the present study, we focused on GLU, GABA, and GLY as the major neurotransmitters in the retina, assessing their concentrations by HPLC in both HD and WT controls at two distinct stages (Appendix A). A mild late-onset imbalance in the excitatory/inhibitory ratio was observed, suggesting that neurochemical dysregulation may emerge at later stages of disease progression (Figure 6A). Glutamate levels were higher at week 10 compared to week 32 in both genotypes. However, no significant differences between HD and WT mice were found at either time points (Figure 6B).

Overall, no significant differences were found in any of the neurotransmitters analysed individually (Appendix A).

### 2.4. Other Retina Neurotransmitters in AD Mice

To complement analysis of the dopaminergic circuit in AD retinas, we assessed the balance between glutamate, GABA, and glycine as key indicators of overall retinal neurotransmission. In the 27-week-old AD mice, GLU levels and the excitatory/inhibitory neurotransmitter ratio showed no significant changes compared to WT controls (Figure 7A), indicating preserved neurotransmitter homeostasis at this early stage. However, in the older AD group, a notable increase in GLU levels was observed (Figure 7B). Moreover, the excitatory/inhibitory neurotransmitter ratio was significantly decreased despite the GLU increase (Figure 7A). This change was primarily driven by increased GABA levels, (Appendix A). This elevation in GABA levels may be explained by previous studies suggesting that Aβ oligomer injection induces GABA accumulation within Müller cells, disrupting normal neurotransmitter balance [72]. These findings underscore the complex neurochemical alterations associated with retinal degeneration in the context of Alzheimer’s disease.

## 3. Discussion

The findings of this study highlight distinct yet overlapping alterations in the retinal dopaminergic circuit (RDC) in Huntington’s disease (HD) and Alzheimer’s disease (AD), reinforcing the potential of the retina as a biomarker for neurodegeneration. Both diseases exhibited dysfunctions in dopamine (DA) metabolism, receptor expression, and neurotransmitter balance, yet they differed in the onset, severity, and compensatory responses of these alterations. Our results show that HD mice present a marked reduction in DA levels as early as 10 weeks, preceding the onset of motor symptoms. This early dysfunction is accompanied by a significant decrease in tyrosine hydroxylase (TH) expression, indicating impaired DA synthesis or DAC neurodegeneration. Notably, a previous study using the R6/2 mouse model reports a reduction in TH mRNA levels without a corresponding decrease in the number of TH-positive cells, indicating that transcriptional downregulation can occur independently of cell loss in some HD models [23].

Moreover, increased DA turnover at later stages suggests an accelerated degradation process that may further contribute to dopaminergic decline. These findings align with previous studies demonstrating that brain DA dysregulation occurs before substantial striatal degeneration in HD, underscoring the potential of RDC dysfunction as an early biomarker of the disease.

In contrast, DA levels in AD mice remained stable for 27 weeks but showed a significant decline at 64 weeks, coinciding with disease progression. Unlike in HD, TH expression in AD mice increased at 64 weeks despite DA reduction, suggesting a compensatory mechanism aimed at counteracting DA loss. This upregulation of TH, however, does not appear to be sufficient to prevent DA depletion, indicating a gradual failure of the dopaminergic system over time. The discrepancy between TH levels and DA availability in AD may reflect impaired DA release or heightened degradation, highlighting a different mechanism of DA dysfunction compared to HD. Another difference between HD and AD is dopamine receptor expression. In HD mice, D1R expression was persistently reduced, contributing to DA signalling deficits. In contrast, AD mice maintained stable D1R levels but showed increased D2R expression at 64 weeks, suggesting an adaptive response to DA depletion. These differences highlight disease-specific trajectories in retinal dopaminergic alterations. To further characterize the impaired DA metabolism in the retinal dopaminergic system, it would be necessary to expand the study to examine the expression of dopamine-degrading enzymes and the DA transporter involved in its reuptake.

Dopamine, while relevant as demonstrated in this study, constitutes a minor neurotransmitter in the retina, which is predominantly regulated by glutamate as the principal excitatory neurotransmitter and GABA or glycine as key inhibitory modulators. In early HD, neurotransmitter levels remain stable, with minor imbalances emerging at later stages. Conversely, Alzheimer’s disease (AD) is characterized by increased glutamate and GABA levels in advanced stages. These findings suggest that HD primarily impacts DA synthesis and receptor expression, whereas AD induces broader neurochemical disruptions extending beyond the late dopaminergic system alterations. However, research from AD patients suggests that the colour contrast, in which the DA pathway plays a pivotal role, is primarily affected [53,67].

Our results show that the RDC, essential for colour contrast perception, is disrupted in both HD and AD, though with different timing and mechanisms. Currently, no standardised ERG protocol exists to assess colour contrast using colour-specific stimuli [73]. Our review and data suggest that developing such a method could support early diagnosis, particularly in HD, where retinal changes appeared before motor symptoms. In contrast, retinal alterations in AD were detected at stages when cognitive decline is already evident. While this study does not propose a clinical tool, it provides preclinical evidence to guide the future development of non-invasive retinal biomarkers for neurodegenerative diseases.

## 4. Materials and Methods

### 4.1. Animal Models, Genotyping, and Retina Collection

The mouse strains included in the study were B6.CgTg(HDexon1)61Gpb/J, also known as R6/1, a model of Huntington’s disease that expresses a fragment of exon 1 of the human HTT gene with an expansion of CAG repeats (~115 repeats), here referred to as HD; and the B6.C3Tg(APPswe/PSEN1ΔE9)85Dbo/Mmjax model that expresses mutated human APP (Swedish mutation) and PSEN1 (ΔE9 deletion), leading to amyloid plaque formation and Alzheimer’s-like pathology, here referred to as AD. B6CBA/OlaHsd was the wild type (WT) control strain for HD, and C57BL/6J for AD. Genotyping was performed postmortem. A fragment of the tail was collected after mice euthanasia, and a High Pure PCR Template Preparation kit (Roche, Basel, Switzerland) was used according to manufacturer’s instructions for DNA isolation. Specific primers for human htt detection in the HD line (Fw: CCGCTCAGGTTCTGCTTTTA, Rv: TGGAAGGACTTGAGGGAC) and for APP in the AD mice (Fw: ATGAGAGAATGGGAAGAG, Rv: CTGGAAATGCTGGATAAC) were used for amplification using AmpliTaq (Roche). Subsequently, a 1.8% or 3% agarose gel electrophoresis was performed to separate amplicons.

Retinas were harvested postmortem following ocular enucleation. A small incision was made below the corneal margin, and the cornea was carefully removed along with the crystalline lens. The retina was then dissected by severing the optic nerve at the level of the optic disc. Isolated retinas were immediately snap-frozen and stored at −80 °C until further processing.

### 4.2. RNA Extraction and qRT-PCR

Left eye retinas were used for RNA extraction with the NZY Total RNA Isolation Kit (NZY Tech) according to the manufacturer’s instructions. After extraction, RNA concentration was measured using the Nanodrop ND-1000 Spectrophotometer (Thermo Fisher Scientific, Waltham, MA, USA). Reverse transcription (RT) was performed with SuperScript II enzyme and using OligodT (Thermo Fisher Scientific) following the manufacturer’s instructions.

For quantitative PCR, Taqman Fast Universal PCR Master Mix (Thermo Fisher Scientific) and the following Taqman mouse probes were used: *Gapdh* (Mm99999915_g1), D1R (Mm02620146_s1), and D2R (Mm00438545_m1), all from Thermo Fisher Scientific. For qPCR, expression data were obtained from standard curve extrapolation and normalized to *Gapdh* mRNA.

### 4.3. High-Performance Liquid Chromatography **(HPLC)**

Retinas from the right eyes were sonicated in 70 µL of 0.4 N perchloric acid for deproteinization. Samples were centrifuged at 12,000 rpm for 30 min at cold temperatures, and the supernatant was collected for analysis. Dopamine (DA) and its metabolites were measured from supernatants by HPLC with an ESA Coulochem detector [74]. The chromatographic conditions were as follows: a column ACE 5 C18, 150 × 4.6 mm; a citrate/acetate buffer 0.1 M, pH 3.9 with 10% methanol, 1 mM EDTA, and 1.2 mM heptane sulfonic acid, flow rate 1 mL/min. Monoamine levels were identified by their retention time and the amounts calculated against calibrated external standard solutions. Amino acids were determined by HPLC, as previously described [75]. Fluorescence detection was accomplished with a Jasco detector (FP-2020) at 240 and 450 nm for excitation and emission wavelengths, respectively. Amino acids were identified by their retention times, and their concentrations were calculated by comparison to calibrated amino acid external standard solutions. The values were adjusted based on the total protein amount, calculated for each retina, and are expressed as picomoles per microgram of protein.

### 4.4. Western Blot

The remaining pellet from HPLC extraction was resuspended in lysis buffer containing 0.75% Na_2_CO_3_, 2% SDS, 1mM PMSF, 20mM β-glycerophosphate, 100 mM NaF, and 20 mM sodium molybdate. The sample was sonicated for 18 s to disaggregate the pellet and then centrifuged at 12,000 rpm for 30 min at 4 °C. The supernatant was collected and stored at −20 °C until processing. Polyacrylamide gels at 10% or 15% were prepared, and 15–20 μg of protein was loaded along with a molecular weight marker. The proteins were separated by electrophoresis and transferred to a nitrocellulose membrane using the Trans-Blot system (BioRad). The membrane was blocked with 5% milk in PBS-Tween 0.1% for 30 min. Primary antibodies were incubated overnight at 4 °C with the following dilutions: anti-TH (1/5000) (Merck, MAB5280), and in a second round, anti-actin (1/10,000) (Sigma Aldrich, A5441, St. Louis, MO, USA) was used for normalization. Secondary antibodies were added after washing: anti-mouse HRP (1/2500) (SouthernBiotech, Birmingham, AL, USA) and anti-rabbit HRP (1/2500) (SouthernBiotech, 4055-05). Detection was carried out by the ChemiDoc system (BioRad, Hercules, CA, USA) using ECL chemiluminescence (BioRad). For western blot analysis, protein band intensities were measured by densitometry and normalized to actin levels.

### 4.5. Statistical Analysis

The groups used in this study were: “HD” for the R6/1 mice, “APP” for the APP/PS1 mice, and “WT” for the B6CBA/OlaHsd and C57BL/6 control mice. Graphs and data analyses were performed using GraphPad Prism software V8. All graphical representations display mean values, with individual data points consistently shown. Error bars represent the standard error of the mean (SEM). Statistical significance was determined using Student’s *t*-test, with significance thresholds set as follows: *p* < 0.05 (*), *p* < 0.01 (**), *p* < 0.005 (***).

## 5. Conclusions

Our study underscores the potential of the retina as a biomarker for neurodegeneration by revealing disease-specific alterations in the retinal dopaminergic circuit (RDC) in HD and AD. While HD is marked by an early decline in dopamine synthesis and receptor expression, AD exhibits a later disruption. Given that RDC dysfunction may precede brain dysfunctions in several neurodegenerative diseases, developing a standardized electroretinography (ERG) method to assess colour contrast perception could enhance early diagnosis and serve as a non-invasive biomarker for disease progression.

## Figures and Tables

**Figure 1 ijms-26-05532-f001:**
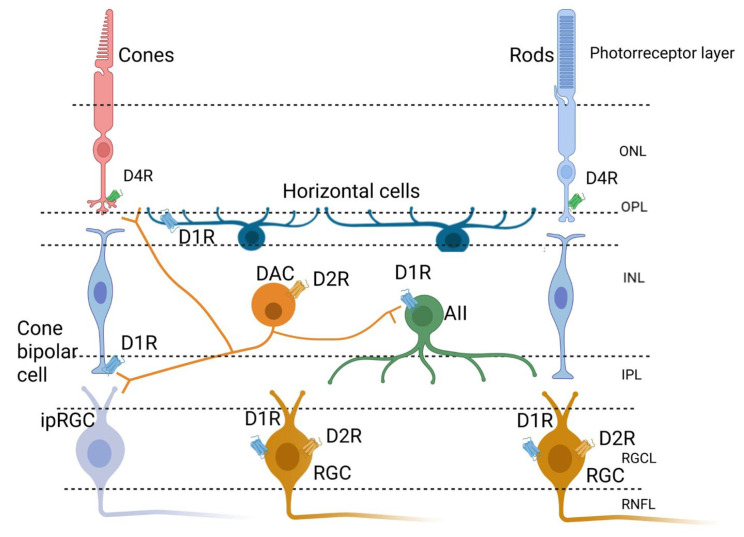
Retinal dopaminergic circuit. The circuit starts with the dopaminergic amacrine cells (DACs) that release dopamine (DA) to the photoreceptors (expressing dopamine receptor 4 (D4R)), to the cone bipolar cells (expressing dopamine receptor 1 (D1R)), to the horizontal cells (expressing D1R), to the amacrine cells AII (expressing D1R), and to the retinal ganglion cells (RGCs, expressing D1R and D2R). DACs also express D2R as an autoreceptor to control the DA release.

**Figure 2 ijms-26-05532-f002:**
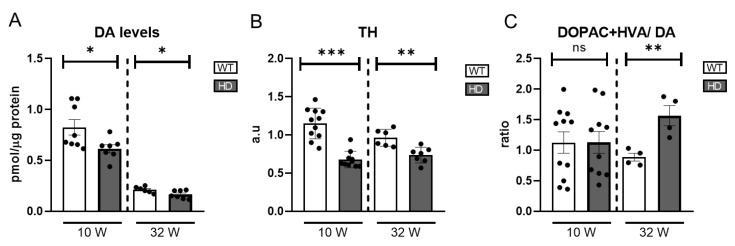
Catecholamines and TH levels in HD mice retinas. (**A**) DA levels were measured by HPLC (n = 7–8). (**B**) TH levels were measured by western blot and normalized to their actin levels. The graph represents the value of bands’ densitometry in arbitrary units (a.u) (n = 6–11). (**C**) Representation of ratio DOPAC+HVA/DA (n = 4–11). ns = non-significant, * = *p* < 0.05, ** = *p* < 0.01, *** = *p* < 0.005.

**Figure 3 ijms-26-05532-f003:**
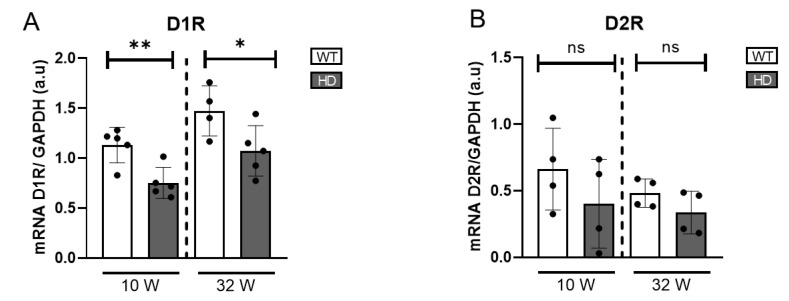
Gene expression of dopamine receptors in HD mice retinas. (**A**) mRNA levels of D1R normalized to GAPDH mRNA (n = 4–5). (**B**) mRNA levels of D2R normalized to GAPDH mRNA (n = 4–5). ns = non significant, * = *p* < 0.05, ** = *p* < 0.01.

**Figure 4 ijms-26-05532-f004:**
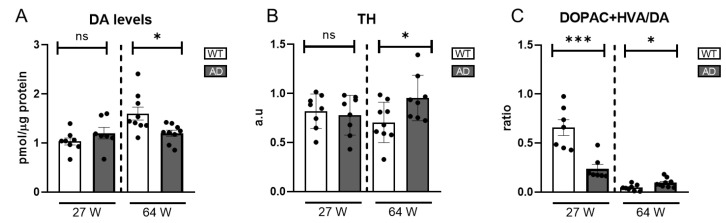
Catecholamines and TH levels in AD mice retinas. (**A**) DA levels were measured by HPLC (n = 7–9). (**B**) Representation of the ratio DOPAC+HVA/DA (n = 8–9). (**C**) TH levels were measured by western blot and the data were normalized to their actin levels (n = 7–9). ns= non-significant, * = *p* < 0.05, *** = *p* < 0.005.

**Figure 5 ijms-26-05532-f005:**
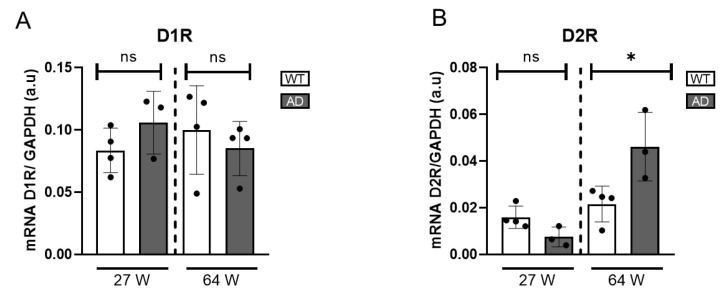
Gene expression of D1R and D2R of AD mice retinas. (**A**) mRNA levels of D1R are shown and normalized to GAPDH mRNA levels (n = 4). (**B**) mRNA levels of D2R are shown and normalized to GAPDH mRNA (n = 3–4). ns = non-significant, * = *p* < 0.05.

**Figure 6 ijms-26-05532-f006:**
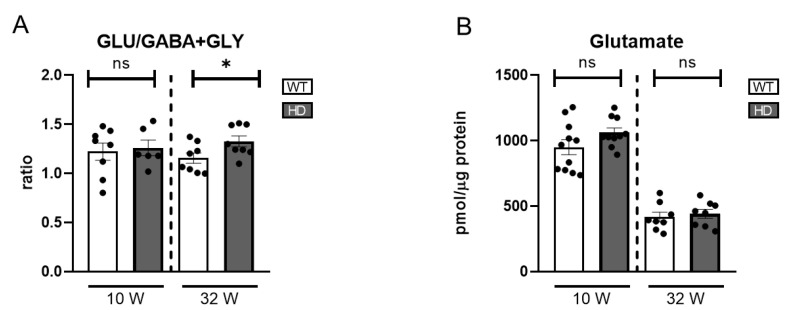
Excitatory and inhibitory neurotransmitter levels in the retinas of HD mice. (**A**) Ratio of glutamate (GLU) to the sum of inhibitory neurotransmitters GABA and glycine (GLY), indicating overall excitatory/inhibitory balance (n = 7–8). (**B**) Absolute levels of glutamate (n = 8–11). ns: no significant differences were found. ns = non-significant, * = *p* < 0.05.

**Figure 7 ijms-26-05532-f007:**
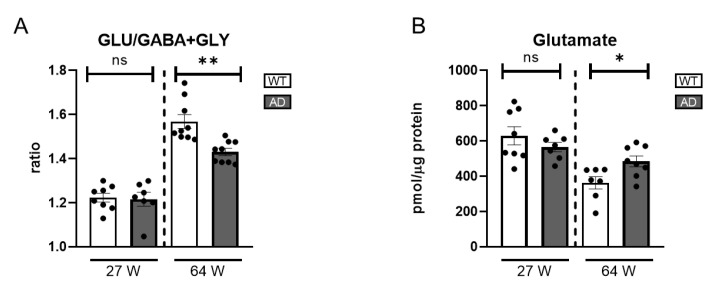
Excitatory and inhibitory neurotransmitter levels in the retinas of AD mice. (**A**) Ratio of glutamate (GLU) to the sum of inhibitory neurotransmitters GABA and glycine (GLY), reflecting the excitatory/inhibitory balance (n = 7–9). (**B**) Absolute glutamate levels (n = 7–8). ns = non-significant; * = *p* < 0.05, ** = *p* < 0.01.

## Data Availability

The raw data supporting the conclusions of this article will be made available by the authors on request.

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
