# Peer review of "The Retinal Dopaminergic Circuit as a Biomarker for Huntington’s and Alzheimer’s Diseases"

_ijms, 2025, doi:10.3390/ijms26125532_

Round 1
Reviewer 1 Report
Comments and Suggestions for Authors
In the manuscript, Blanco-Hernán et al. investigated the alterations in the retinal dopaminergic circuit (RDC) in an attempt to establish such alterations as biomarkers for Huntington’s (HD) and Alzheimer’s diseases (AD). Specifically, the authors assessed the dopamine (DA) levels and metabolism, tyrosine hydroxylase (TH) expression, dopamine receptors (D1R and D2R) gene expression, excitatory/inhibitory neurotransmitters, and inflammatory markers in murine models of HD and AD. Significant RDC disruptions preceding brain degeneration and motor symptoms in the HD murine model were observed but not in the AD murine model. As a result, the authors proposed that the retinal biomarkers may serve as promising biomarkers for the neurodegenerative diseases.
The study is a valuable exploration toward the early stage diagnosis of HD and AD, and thus is important and potentially valuable.
However, several major issues are present in the manuscript and need to be addressed before it can be considered further.
- The manuscript is somewhat ambiguous and confusing. The title indicates that the manuscript will investigate the potential of RDC as a biomarker for HD and AD. However, in the Abstract, the authors stated that “the retina is increasingly recognized as a potential biomarker for neurodegenerative diseases” and “this study highlights the value of retinal biomarkers in neurodegenerative research”. In fact, what the authors actually investigated was retinal dysfunction; in other words, retinal dysfunction may be an early marker in the progression of AD and HD. It seems that retinal dysfunction refers to the alterations in RDC. However, the authors did not provide a clear explanation for the definition of RDC (Figure 1 shows the dopaminergic retinal circuit, is this identical to retinal dopaminergic circuit?). Taken together, it appears that the authors used multiple different but closely-related concepts in the manuscript and caused great confusion.
To avoid confusion, it may be better and clearer to focus the investigations in the manuscript on retinal dysfunction rather than alterations in RDC. If the authors insist on using the concept of RDC, they must first define the concept clearly and indicate the elements of RDC.
- A contradiction is present in the manuscript. In the Introduction, the authors stated that “assessing this circuit directly or evaluating its role in colour contrast regulation and light sensitivity may offer a novel approach for early disease detection (L232-233)”, and in the Abstract, it was stated “due to its accessibility, retinal assessment via Electroretinography or Optical Coherence Tomography emerges as a promising non-invasive method for early diagnosis and ongoing monitoring” (L24-25). These statements give readers an impression that the manuscript will develop a non-invasive method for early diagnosis of HD and AD. However, the authors investigated dopamine levels and metabolism, tyrosine hydroxylase expression, dopamine receptors gene expression, excitatory/inhibitory neurotransmitters, and inflammatory markers, all of which are invasive methods. In fact, these endpoints can only be assessed in the animals. How can one make early diagnosis of HD and AD in humans through the biomarkers?
- The markers measured in the manuscript include the dopamine (DA) levels and metabolism, tyrosine hydroxylase (TH) expression, dopamine receptors (D1R and D2R) gene expression, excitatory/inhibitory neurotransmitters, and inflammatory markers (L14-16), among which excitatory/inhibitory neurotransmitters and inflammatory markers are not indicated specifically. Because there are many different excitatory/inhibitory neurotransmitters and inflammatory markers, these markers should be indicated specifically (e.g., glutamate, GABA, glycine, etc. Fig. 4).
- Organization of the manuscript is somewhat confusing. The manuscript first presented the data of the levels of DA and TH, DA degradation metabolites, D1R/D2R, and GLU/GABA/Gly in the HD murine model (Sections 2.1-2.2), and then such data in the AD murine model (Sections 2.3-2.4). However, the data of inflammatory markers are presented in Section 2.5 for both models, which is odd. It may be better to present the data in the same sections side by side for two murine models to facilitate comparisons between the two models.
- Some experimental details are missing, including animal treatment, collection of retina, measurement of Iba-1. etc.
- Writing of the manuscript needs a major improvement. The major problem in the writing is that the Introduction is simply too long. Initially, it gives readers an impression that the manuscript is a review rather than a research article. Clearly, the Introduction must be shortened greatly.
Author Response
In the manuscript, Blanco-Hernán et al. investigated the alterations in the retinal dopaminergic circuit (RDC) in an attempt to establish such alterations as biomarkers for Huntington’s (HD) and Alzheimer’s diseases (AD). Specifically, the authors assessed the dopamine (DA) levels and metabolism, tyrosine hydroxylase (TH) expression, dopamine receptors (D1R and D2R) gene expression, excitatory/inhibitory neurotransmitters, and inflammatory markers in murine models of HD and AD. Significant RDC disruptions preceding brain degeneration and motor symptoms in the HD murine model were observed but not in the AD murine model. As a result, the authors proposed that the retinal biomarkers may serve as promising biomarkers for the neurodegenerative diseases.
The study is a valuable exploration toward the early-stage diagnosis of HD and AD, and thus is important and potentially valuable.
However, several major issues are present in the manuscript and need to be addressed before it can be considered further.
Response: we thank the reviewer for the constructive feedback and recognition of the work’s relevance. We have addressed all major concerns and provided clear explanations for the changes made:
Comment 1: The manuscript is somewhat ambiguous and confusing. The title indicates that the manuscript will investigate the potential of RDC as a biomarker for HD and AD. However, in the Abstract, the authors stated that “the retina is increasingly recognized as a potential biomarker for neurodegenerative diseases” and “this study highlights the value of retinal biomarkers in neurodegenerative research”. In fact, what the authors actually investigated was retinal dysfunction; in other words, retinal dysfunction may be an early marker in the progression of AD and HD. It seems that retinal dysfunction refers to the alterations in RDC. However, the authors did not provide a clear explanation for the definition of RDC (Figure 1 shows the dopaminergic retinal circuit, is this identical to retinal dopaminergic circuit?). Taken together, it appears that the authors used multiple different but closely-related concepts in the manuscript and caused great confusion. To avoid confusion, it may be better and clearer to focus the investigations in the manuscript on retinal dysfunction rather than alterations in RDC. If the authors insist on using the concept of RDC, they must first define the concept clearly and indicate the elements of RDC.
Response 1 : a revised version of the abstract has been written to address the reviewer’s concerns. Specifically, we have clarified that the study focuses on dysfunction within the Retinal Dopaminergic Circuit (RDC) as a potential early marker of neurodegeneration, rather than suggesting the retina as a whole serves as a biomarker. We have also provided a clearer definition of the RDC and explained its functional relevance within the visual system. These revisions ensure that the abstract is now fully aligned with the manuscript title. We hope these changes resolve the previous confusion and improve the overall clarity and focus of the work.
Comment 2: a contradiction is present in the manuscript. In the Introduction, the authors stated that “assessing this circuit directly or evaluating its role in colour contrast regulation and light sensitivity may offer a novel approach for early disease detection (L232-233)”, and in the Abstract, it was stated “due to its accessibility, retinal assessment via Electroretinography or Optical Coherence Tomography emerges as a promising non-invasive method for early diagnosis and ongoing monitoring” (L24-25). These statements give readers an impression that the manuscript will develop a non-invasive method for early diagnosis of HD and AD. However, the authors investigated dopamine levels and metabolism, tyrosine hydroxylase expression, dopamine receptors gene expression, excitatory/inhibitory neurotransmitters, and inflammatory markers, all of which are invasive methods. In fact, these endpoints can only be assessed in the animals. How can one make early diagnosis of HD and AD in humans through the biomarkers?
Response 2: we appreciate the reviewer’s observation about the apparent contradiction between the methods used in this study and the statements in the Introduction and Abstract regarding early diagnosis through non-invasive approaches. We agree that our experimental endpoints are indeed invasive and limited to animal models. However, the purpose of these experiments was to characterize in detail the dysfunction of the retinal dopaminergic circuit (RDC) in neurodegeneration, with the goal of identifying physiological processes that could later be assessed through non-invasive techniques in humans. As discussed in the manuscript (see abstract, introduction lines 50-53 and discussion, lines 441–446), we specifically propose that electroretinography (ERG), particularly if optimized to evaluate color contrast perception, could be developed as a non-invasive method for detecting RDC dysfunction. Although such a protocol is not yet standardized, our findings highlight the biological basis for its potential diagnostic value, especially in Huntington’s disease, where retinal changes prece de motor symptoms.
To address this concern and eliminate any confusion, we have revised the relevant sections of the Abstract and Introduction to clarify that our current work provides preclinical evidence supporting the development of future non-invasive diagnostic tools, rather than presenting or validating such tools directly in this study. We hope this clarification resolves the concern.
Comment 3: the markers measured in the manuscript include the dopamine (DA) levels and metabolism, tyrosine hydroxylase (TH) expression, dopamine receptors (D1R and D2R) gene expression, excitatory/inhibitory neurotransmitters, and inflammatory markers (L14-16), among which excitatory/inhibitory neurotransmitters and inflammatory markers are not indicated specifically. Because there are many different excitatory/inhibitory neurotransmitters and inflammatory markers, these markers should be indicated specifically (e.g., glutamate, GABA, glycine, etc. Fig. 4).
Response 3: in response to both reviewers in this matter, and to maintain a focused narrative around the RDC as a potential early biomarker, we have removed the section related to inflammatory markers from the manuscript. Although inflammation is relevant in neurodegeneration, the results obtained were not novel in the context of retinal studies and would have diluted the main message of the paper.
Regarding the excitatory/inhibitory neurotransmitters, we agree that clarification is needed. In the retina, the range of neurotransmitters is more limited compared to the brain. The primary excitatory neurotransmitter is glutamate (GLU), while the main inhibitory ones are GABA and glycine (Gly). Our aim was to determine whether dopaminergic dysfunction influenced the balance between these major neurotransmitters. We have now specified this clearly in the Results sections, and the revised version of Figure 4 highlights the individual measurements of GLU, GABA, and Gly.
Comment 4: organization of the manuscript is somewhat confusing. The manuscript first presented the data of the levels of DA and TH, DA degradation metabolites, D1R/D2R, and GLU/GABA/Gly in the HD murine model (Sections 2.1-2.2), and then such data in the AD murine model (Sections 2.3-2.4). However, the data of inflammatory markers are presented in Section 2.5 for both models, which is odd. It may be better to present the data in the same sections side by side for two murine models to facilitate comparisons between the two models.
Response 4: We appreciate the reviewer’s suggestion regarding the manuscript structure. Originally, we opted for a sequential presentation (first HD, then AD) to reflect the timeline of our experimental work and to preserve clarity within each model. However, we agree that presenting the data in a side-by-side format would facilitate direct comparison. In the revised version, we have reorganized the Results section accordingly, aligning the data for HD and AD models in each subsection (e.g., DA and TH, receptor expression, neurotransmitter balance). Additionally, since we ultimately decided to remove the inflammatory marker data to maintain a focused narrative around RDC dysfunction, this concern regarding the placement of inflammation data is no longer applicable.
Comment 5: some experimental details are missing, including animal treatment, collection of retina, measurement of Iba-1. etc.
Response 5: additional experimental details have now been included in the revised Methods section. Lines 465-469
Comment 6: writing of the manuscript needs a major improvement. The major problem in the writing is that the Introduction is simply too long. Initially, it gives readers an impression that the manuscript is a review rather than a research article. Clearly, the Introduction must be shortened greatly.
Response 6: we fully agree that the original Introduction was too long and risked giving the impression of a review article. However, since this is the first study to propose a specific retinal circuit (retinal dopaminergic circuit) as a candidate biomarker in neurodegeneration, we thought it was important to provide a concise but solid review of the supporting literature. That said, in response to both your comment and similar feedback from Reviewer #2, we have substantially revised and shortened the Introduction. The revised version retains only the essential background needed to contextualize the study.
Reviewer 2 Report
Comments and Suggestions for Authors
Dear Editor,
The manuscript discusses changes in the retinal dopaminergic system in mouse models of Huntington’s disease (HD) and Alzheimer’s disease (AD). The authors use HPLC, immunoblotting, and qRT-PCR to measure dopamine (DA) levels and its related metabolites. While the study attempts to compare two neurodegenerative disease models, the presentation is somewhat confusing. It would be more effective to focus the results on one disease model for clarity and depth.
Major Concerns:
- The comparison between HD and AD models is not clearly justified and makes it difficult for the reader to follow. I recommend narrowing the focus to a single model to strengthen the analysis and interpretation of the results.
- The manuscript does not provide a clear explanation for why the observed changes occur at certain stages or in specific disease conditions. More mechanistic studies are needed to understand the causes of retinal changes in these models.
- The introduction is too long and reads more like a review article. It should be shortened and revised to guide readers directly to the research question and the rationale behind the study.
- The reported changes in GFAP and Iba1 levels in HD retina are not novel. A recent study by F. Cano-Cano et al. titled “Retinal dysfunction in Huntington’s disease mouse models concurs with local gliosis and microglia activation” has already reported similar findings in R6/1 mice. Please cite this study and clarify how your findings add new information.
- The current data rely heavily on western blots, which do not provide information about cell location, cell numbers, or cell death. Please include immunostained retinal images for markers such as TH, dopamine receptors, GFAP, and Iba1 for better visualization and interpretation.
- Since there is an increase in GFAP and Iba1, did the authors examine other inflammatory markers? Additional analysis would help clarify the extent of inflammation in the retina.
Minor Concerns:
- The molecular weight of actin is incorrectly labeled as 55 kDa. It should be corrected to 42 kDa.
- Figure 1 is too large and overlaps with the text at line 253. Please resize the figure for better layout and readability.
Author Response
Comment 1-The comparison between HD and AD models is not clearly justified and makes it difficult for the reader to follow. I recommend narrowing the focus to a single model to strengthen the analysis and interpretation of the results.
Response 1-Our rationale for comparing these two conditions is based on their distinct etiologies (genetic versus sporadic) and their known but differential involvement of dopaminergic systems. Since this is the first study to propose a specific retinal circuit (the RDC) as a site of early vulnerability in neurodegeneration, we aimed to determine whether RDC dysfunction is a shared or disease-specific feature. We agree that clarity is essential, and in the revised manuscript we have improved the structure and explicitly framed the comparison as exploratory, focusing on how RDC impairment manifests differently in each model ( see new abstract and introduction lines 47 to 52).
Comment 2- The manuscript does not provide a clear explanation for why the observed changes occur at certain stages or in specific disease conditions. More mechanistic studies are needed to understand the causes of retinal changes in these models.
Response 2- We acknowledge the reviewer’s request for deeper mechanistic insight. However, the main objective of this study was not to elucidate upstream molecular mechanisms, but rather to characterize early alterations in the RDC and evaluate its potential as a biomarker for neurodegenerative diseases. Given that this work was submitted to a Special Issue focused on biomarkers in neurodegeneration, our intention was to contribute novel, circuit-level evidence from the retina that could guide the development of future diagnostic strategies.
Importantly, this is the first report to suggest that a specific retinal circuit may be selectively vulnerable in neurodegeneration. By identifying consistent early alterations in RDC components (dopamine levels, TH expression or receptor gene expression) we aim to provide a foundation for future mechanistic studies. We have now clarified this rationale more explicitly in the revised manuscript and expanded our discussion of how disease stage and pathology may influence the timing and pattern of retinal changes observed in HD versus AD models.
Comment 3- The introduction is too long and reads more like a review article. It should be shortened and revised to guide readers directly to the research question and the rationale behind the study.
Response 3- We agree with the reviewer that the original Introduction was too long and included excessive background detail. However, as this is the first study to propose the RDC as a target in neurodegeneration, we felt it was necessary to include a brief but comprehensive review of existing retinal findings in HD and AD to justify the rationale behind our hypothesis. In response to the reviewer’s comments, we have significantly revised and shortened the Introduction to make it more direct and focused, while retaining key references that support the relevance and plausibility of our research question
Comment 4-The reported changes in GFAP and Iba1 levels in HD retina are not novel. A recent study by F. Cano-Cano et al. titled “Retinal dysfunction in Huntington’s disease mouse models concurs with local gliosis and microglia activation” has already reported similar findings in R6/1 mice. Please cite this study and clarify how your findings add new information.
Response 4- We were aware of this paper and have cited it several times but regarding inflammation please find an answer to this issue in comment 6
Comment 5-The current data rely heavily on western blots, which do not provide information about cell location, cell numbers, or cell death. Please include immunostained retinal images for markers such as TH, dopamine receptors, GFAP, and Iba1 for better visualization and interpretation.
Response 5- We agree with the reviewer regarding the value of immunohistochemistry for visualizing cellular localization and quantifying changes in cell populations. However, for the markers addressed in the manuscript, there are important limitations or considerations: First, reliable antibodies for D1 and D2 dopamine receptors suitable for immunohistochemistry are currently not available, as confirmed in previous studies and our own experience. For this reason, receptors expression was assessed at the transcript level. Second, regarding tyrosine hydroxylase (TH), while we agree that spatial localization is valuable, it is well established that TH expression in the retina is restricted to a small subset of amacrine cells. These cells are sparse and distributed across the inner retina, making their quantification in histological sections challenging and potentially unreliable. Moreover, TH levels can be altered without actual loss of dopaminergic amacrine cells, affecting dopamine synthesis capacity despite their presence. For this reason, we consider total TH protein quantification by Western blot in retinal lysates a more robust and informative readout of the retina’s dopaminergic potential in this context
Comment 6- Since there is an increase in GFAP and Iba1, did the authors examine other inflammatory markers? Additional analysis would help clarify the extent of inflammation in the retina.
Response 6- We thank the reviewer for this suggestion. In further studies, we might explore additional inflammatory markers beyond GFAP and Iba1. However, in response to reviewer 1 as well, we have decided to remove this part from the final manuscript. Our rationale was to maintain a clear focus on the retinal dopaminergic circuit (RDC) as a potential early and specific biomarker for neurodegenerative diseases.
As you kindly pointed out the inflammatory changes observed are consistent with previously reported findings in neurodegenerative models and, although relevant, do not add novelty or mechanistic insight in the context of RDC-specific dysfunction. Therefore, we believe that expanding the analysis to additional inflammatory markers would fall outside the scope of this work and would detract from the main objective, which is to characterize early RDC alterations that may be used to inform future non-invasive diagnostic strategies.
Minor Concerns:
- The molecular weight of actin is incorrectly labeled as 55 kDa. It should be corrected to 42 kDa.
Corrected
- Figure 1 is too large and overlaps with the text at line 253. Please resize the figure for better layout and readability.
Corrected
Round 2
Reviewer 1 Report
Comments and Suggestions for Authors
The issues raised for the original manuscript have been addressed and the revised manuscript can be accepted for publication. However, some minor text-editing errors need to be corrected. Some examples are listed as follows.
L54. A full stop is missing.
L56. HD can be directly used because it has been defined before (L46). It is similar for AD (L97).
L72 and elsewhere. “supplementary table 1”: Supplementary Table 1.
L105 and 123. A comma should be deleted.
L108. A space should be added.
L123. “where found”: have been found.
L130. A space should be deleted.
Author Response
Dear Reviewer,
Thank you very much for your positive assessment of our revised manuscript and for recommending it for publication. We greatly appreciate your careful reading and your helpful comments.
We have implemented all the minor text-editing corrections you pointed out and have carefully reviewed the manuscript to ensure consistency and clarity throughout.
Thank you again for your valuable feedback and support.
Reviewer 2 Report
Comments and Suggestions for Authors
Dear Editor,
Thank you for sharing the revised version of the manuscript titled “The Retinal Dopaminergic Circuit as a Biomarker for Huntington’s and Alzheimer’s Diseases” by Blanco-Hernan et al. The authors have made notable improvements, especially in the Introduction section, which is much clearer now.
However, there are still a few areas that need further attention:
- Please ensure that the animal models are labeled consistently and correctly throughout the manuscript. For example, the Huntington’s disease model is often referred to as “HD” instead of “R6/1,” while the Alzheimer’s disease model is labeled as “APP.” It would be helpful to use the appropriate and consistent terminology for both.
- The number of animals used is not clearly stated in several figures. Kindly include the sample size for all experiments to ensure transparency and reproducibility.
- In response to point 5, while I understand that TH expression can vary due to multiple factors, many published studies have successfully shown TH staining in the mouse retina. It is still unclear why the authors have chosen not to include this data. The same concern applies to GFAP and Iba1 staining. Providing this data would strengthen the manuscript.
- Please include the full, uncropped blots for all western blot experiments to support the presented results.
Thank you.
Author Response
Dear Editor,
Thank you for your thoughtful and constructive feedback on our revised manuscript. We appreciate the reviewer’s recognition of the improvements, particularly in the Introduction, and their guidance on the remaining issues. Please find our responses to reviewer 2 below:
- Please ensure that the animal models are labeled consistently and correctly throughout the manuscript. For example, the Huntington’s disease model is often referred to as “HD” instead of “R6/1,” while the Alzheimer’s disease model is labeled as “APP.” It would be helpful to use the appropriate and consistent terminology for both.
We have carefully revised the manuscript to ensure consistent and accurate use of terminology for both models. We now refer to the Huntington’s disease model as “HD” and the Alzheimer’s disease model as “AD” throughout the text, figure legends, and tables. We have carefully explained the complete genotype and origin of the models in Material and Methods
- The number of animals used is not clearly stated in several figures. Kindly include the sample size for all experiments to ensure transparency and reproducibility
We have updated all relevant figure legends to clearly indicate the number of animals used in each experiment. Each data point in the scatter plots represents a single retina from an individual mouse. But the number has now been explicitly stated to ensure clarity, transparency, and reproducibility.
- In response to point 5, while I understand that TH expression can vary due to multiple factors, many published studies have successfully shown TH staining in the mouse retina. It is still unclear why the authors have chosen not to include this data. The same concern applies to GFAP and Iba1 staining. Providing this data would strengthen the manuscript.
We sincerely appreciate the reviewer’s insightful comments and fully acknowledge the relevance of TH as a marker for identifying retinal dopaminergic neurons. However, in our experience, TH-positive amacrine cells are very sparsely and inconsistently distributed across retinal sections. This variability makes section-based quantification unreliable and potentially misleading as a measure of dopaminergic activity in the whole retina. Moreover, a previous study by Ouk et al in 2016 in whole mount retinas from the R6/2 mouse model reported a reduction in TH mRNA levels without the corresponding decrease in the number of TH-positive cells (this is now included in the discussion).
Based on this rationale, we did not perform TH immunostaining across the different models and time points included in the study. Generating and processing the required slides or whole mount retinas at those stages would now involve a timeline exceeding one year, which is unfortunately not feasible at this stage of the project.
To support our reasoning, we included you a representative image of a TH-positive amacrine cell in a wild-type retina perform in our lab. As the image shows, some sections contain a single labeled cell, others show none, and it is rare to observe more than one per section. ( see in the letter attached)
Regarding GFAP and Iba1, we opted to exclude these data from the revised manuscript because their expression in retinal gliosis has been extensively documented in both Huntington’s and Alzheimer’s models. Given their limited novelty and our focus on dopaminergic signaling as a potential biomarker, we believe their inclusion would not substantially strengthen the current study.
We hope this explanation clarifies our choices and we are grateful for the opportunity to address these important points.
- Please include the full, uncropped blots for all western blot experiments to support the presented results.
We have now included a file with full, uncropped versions of all western blot images ready to include in supplementary material requested by the journal.
Thank you once again for your valuable feedback and for the opportunity to further improve our work.
Sincerely,
Dra Mansilla on behalf of all co-authors

Round 3
Reviewer 2 Report
Comments and Suggestions for Authors
Dear Editor,
Thank you for sharing the revised version of the manuscript titled "The Retinal Dopaminergic Circuit as a Biomarker for Huntington’s and Alzheimer’s Diseases" by Blanco-Hernán et al.
The authors have addressed most of my previous comments. However, there are still serious concerns regarding the supplementary figures:
- According to the text, the samples are from WT and HD mice at 10 and 32 weeks. However, the supplementary figures show data for 27 and 64 weeks. This inconsistency raises concerns about the reliability of the data. It also contradicts the authors’ claim that they have carefully reviewed the manuscript. Such carelessness suggests a lack of attention to detail and seriousness in correcting earlier mistakes.
- The supplementary file does not include any uncropped full blots. These are essential for transparency and proper review. Please ensure that full, uncropped versions of all blots are included in the supplementary materials.
Author Response
Dear Reviewer Thank you for your continued review of our revised manuscript. We would like to clarify the issue regarding the mismatch between the ages mentioned in the manuscript (10 and 32 weeks) and those appearing in the supplementary figures (27 and 64 weeks). This was a labeling error: the titles from AD model data figures (which include 27 and 64-week-old animals) were mistakenly dragged into the HD figure panels during file preparation. The underlying data shown in the supplementary figures correspond correctly to HD and WT mice at 10 and 32 weeks. This mistake does not affect the results or conclusions, but we acknowledge the oversight and have corrected the labels accordingly. As for the uncropped Western blots, we had submitted them previously as a separate document, assuming that the inclusion of raw data in the formal supplementary material would be left to the editor’s discretion. However, it is possible that the file was not uploaded correctly or received as intended. We apologize for any confusion and are now re-attaching the full, uncropped blots as a separate document to ensure transparency and facilitate proper review. We appreciate your attention to these details and remain committed to maintaining the integrity and clarity of our work.
Round 4
Reviewer 2 Report
Comments and Suggestions for Authors
Dear Editor,
Thank you for sending the revised version of the manuscript. I sincerely appreciate the authors' efforts in addressing the comments and revising the manuscript. I have reviewed the uncropped blots, and they appear convincing. I have no further comments.
Thank you.